# A protein secreted by the *Salmonella* type III secretion system controls needle filament assembly

**Junya Kato[1†], Supratim Dey[2†], Jose E Soto[1†], Carmen Butan[1], Mason C Wilkinson[2], Roberto N De Guzman[2]*, Jorge E Galan[1]***

[1]Department of Microbial Pathogenesis, Yale University School of Medicine, New Haven, United States; [2]Department of Molecular Biosciences, University of Kansas, Lawrence, United States

**Abstract** Type III protein secretion systems (T3SS) are encoded by several pathogenic or symbiotic bacteria. The central component of this nanomachine is the needle complex. Here we show in a *Salmonella* Typhimurium T3SS that assembly of the needle filament of this structure requires OrgC, a protein encoded within the T3SS gene cluster. Absence of OrgC results in significantly reduced number of needle substructures but does not affect needle length. We show that OrgC is secreted by the T3SS and that exogenous addition of OrgC can complement a Δ*orgC* mutation. We also show that OrgC interacts with the needle filament subunit PrgI and accelerates its polymerization into filaments in vitro. The structure of OrgC shows a novel fold with a shared topology with a domain from flagellar capping proteins. These findings identify a novel component of T3SS and provide new insight into the assembly of the type III secretion machine.
DOI: https://doi.org/10.7554/eLife.35886.001

***For correspondence:**
rdguzman@ku.edu (RNDG);
jorge.galan@yale.edu (JEG)

[†]These authors contributed equally to this work

**Competing interests:** The authors declare that no competing interests exist.

## Introduction

Type III protein secretion systems (T3SSs) are highly conserved molecular machines encoded by many gram-negative bacteria pathogenic or symbiotic to animals, plants, or insects (*Galán et al., 2014*; *Deng et al., 2017*; *Notti and Stebbins, 2016*). These systems are evolutionarily related to flagella, sharing many elements of the machinery that mediates the assembly of this complex organelle (*Diepold and Armitage, 2015*). Unlike flagella, which have evolved to propel bacteria through liquid environments, T3SS machines have evolved to deliver bacterial effector proteins into eukaryotic cells to modulate cellular functions, thus shaping the functional interface between symbionts or pathogens and their hosts (*Galán, 2009*). The entire T3SS machine or injectisome is composed of the needle complex (NC), which is embedded in the bacterial envelope (*Kubori et al., 1998*), and a large cytoplasmic structure known as the sorting platform (*Lara-Tejero et al., 2011*). The NC consists of a cylindrical base ~26 nm in diameter and ~32 nm in height, which is anchored to the inner and outer membranes through multiple ring-shaped structures, and a needle-like appendage or filament that protrudes several nanometers from the bacterial surface (*Marlovits et al., 2004*; *Schraidt et al., 2010*; *Worrall et al., 2016*). The cytoplasmic sorting platform consists of a six-pod structure 23 nm in height and 36 nm in width, capped at one of its ends by a six-spoke, wheel-like structure, all together arranged in a cage-like assembly (*Hu et al., 2017*). The building of this complex organelle occurs in a highly organized fashion that is initiated by the step-wise assembly of the NC, followed by the formation of the cytoplasmic sorting platform (*Sukhan et al., 2001*; *Diepold and Wagner, 2014*). The assembly of the NC is initiated by the organization of a multi-protein membrane complex known as the export apparatus (*Wagner et al., 2010*). This complex templates the assembly of the inner rings of the NC base, which are subsequently linked to the independently assembled outer

ring. Once the base and sorting platform are assembled, the intermediate substructure becomes competent for the secretion of 'early substrates', which are those necessary for the assembly of the inner rod and needle filament. The mechanisms of assembly of the inner rod and needle substructures are incompletely understood. The needle is built by addition of its single subunit at the growing tip after transiting through the central channel of the nascent filament (*Poyraz et al., 2010*). Much less is known about the structure and assembly of the inner rod, which is also formed by a single subunit and links the needle substructure to the base (*Marlovits et al., 2004*; *Lefebre and Galán, 2014*). Here we describe a hitherto uncharacterized component of T3SSs, OrgC, which is required for the initiation of the assembly of the needle substructure of a *Salmonella enterica* serovar Typhimurium (*S.* Typhimurium) T3SS.

## Results

### OrgC is an early substrate of the *S. Typhimurium* T3SS encoded within its pathogenicity island 1

OrgC (oxygen regulated gene C) is encoded within and co-regulated with the cluster of genes that encode the core components of the *S.* Typhimurium T3SS located within the pathogenicity island 1 (SPI-1) (*Figure 1A*) (*Jones and Falkow, 1994*; *Klein et al., 2000*). Previous studies have shown that OrgC is secreted by the SPI-1 T3SS and since it was reported to be non-essential for type III secretion function, it was hypothesized to be an effector protein of this system (*Aguirre et al., 2006*; *Day and Lee, 2003*). However, its translocation into host cells could not be detected (*Day and Lee, 2003*). Homologs of OrgC encoded within T3SS gene clusters with high degree of synteny can be detected in several bacterial species (*Figure 1B* and *Figure 1—figure supplement 1*), suggesting a potentially conserved function in type III secretion.

In an effort to clarify the function of OrgC, we investigated its secretion in specific T3SS mutant backgrounds. Type III secretion occurs in a hierarchical manner and the order in which a protein is engaged by the secretion machine has predictive value about its potential function (*Galán et al., 2014*). Proteins involved in the assembly of the secretion machine are engaged first (early substrates), followed by the proteins translocases (middle substrates), and finally the effector proteins (late substrates). In the *S.* Typhimurium SPI-1 T3SS at least two regulatory proteins, InvJ and SpaS, are involved in regulating the 'substrate switching' of the secretion machine from early to middle and late substrates (*Kubori et al., 2000*; *Monjarás Feria et al., 2015*). Absence of InvJ results in secretion machines 'locked' in the early substrate mode thus leading to long needles and inability to secrete middle and late substrates (*Kubori et al., 2000*). Similarly, a mutation in the catalytic site of the export apparatus component SpaS ($spaS^{N258A}$) results in a secretion machine unable to secrete middle and late substrates (*Monjarás Feria et al., 2015*). As previously reported (*Day and Lee, 2003*), we found that OrgC is secreted into the *S.* Typhimurium culture supernatant in a SPI-1 T3SS-dependent manner (*Figure 1C*). Importantly, we found that OrgC is also secreted in the Δ*invJ* (*Figure 1C*) or $spaS^{N258A}$ (*Figure 1D*) mutant strains, indicating that it is an early substrate of the SPI-1 T3SS. As early substrates are not injected into target host cells, these observations are consistent with previous reports failed to detect the translocation of OrgC into mammalian cells (*Day and Lee, 2003*), and suggest that it may be involved in the assembly of the secretion machine.

### OrgC is required for efficient needle assembly

We investigated the secretion of early, late and middle substrates in a Δ*orgC S.* Typhimurium mutant strain. We found that the Δ*orgC* mutant exhibited reduced secretion of middle (i. e. SipB) and late (i. e. SptP) substrates (*Figure 2A and B*). Notably, the Δ*orgC* mutant exhibited significantly increased level of secretion of the needle and inner rod subunit proteins PrgI and PrgJ. These observations suggest that the Δ*orgC* mutant must have a defect in needle or inner rod assembly because secretion of middle and late substrates requires substrate switching, which can only occur upon completion of the needle filament and its subsequent attachment to the inner rod (*Lefebre and Galán, 2014*; *Marlovits et al., 2006*). The presence of increased levels of needle and inner rod subunits in the culture supernatant, presumably the result of their non-productive secretion, is consistent with this hypothesis.

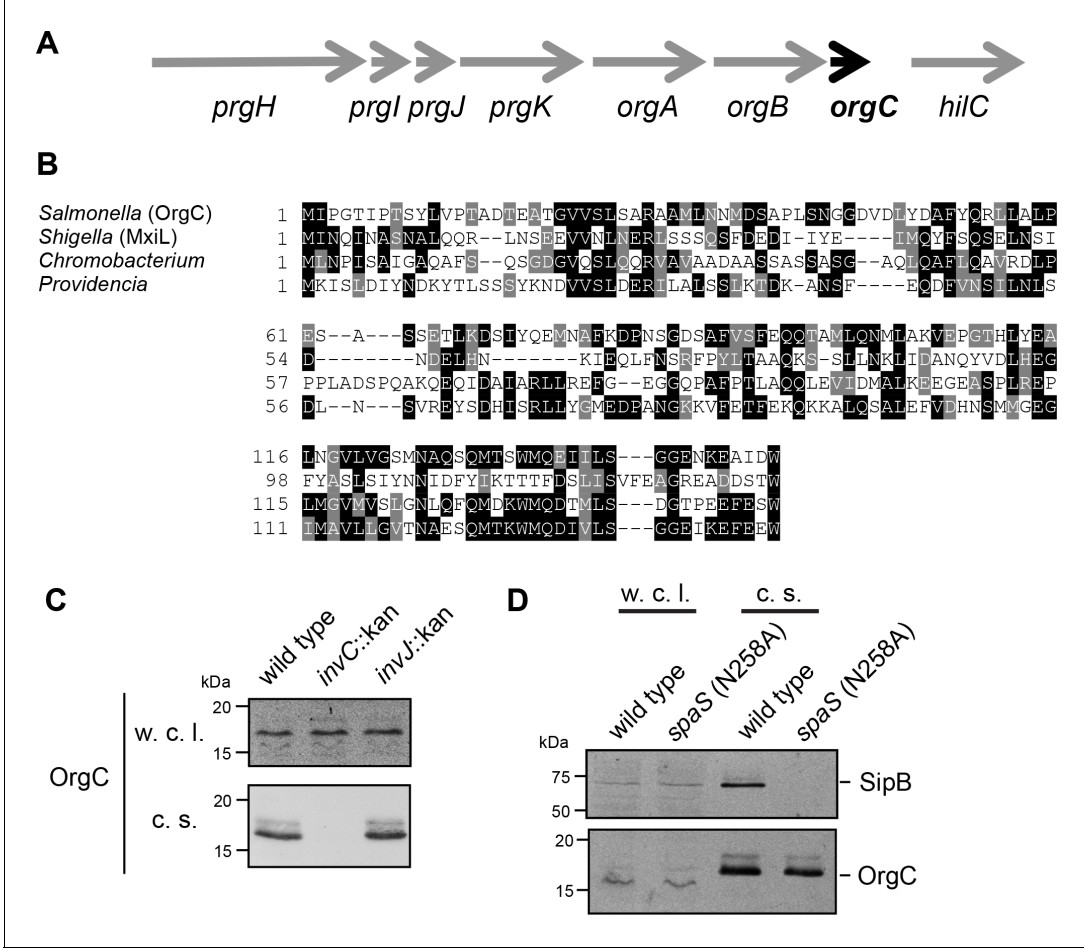

**Figure 1.** The T3SS-associated protein OrgC is secreted as an early substrate of the *S.* Typhimurium type III secretion system encoded within its pathogenicity island 1. (**A**) Gene organization of the *orgC* locus within the *S.* Typhimurium pathogenicity island 1. (**B**) Amino acid sequence alignment of OrgC homologs. The sequences used in the alignment are: OrgC (*S.* Typhymurium), MxiL (Shigella flexneri), and hypothetical proteins from *Chromobacterium violaceum* and *Providencia alcalifaciens*. (**C**) Whole cell lysates (w. c. l.) or culture supernatants (c. s.) of wild-type *S.* Typhimurium, or the isogenic mutants ΔinvC (T3SS-defective), ΔinvJ, or spaS$^{N258A}$ (both mutant strains are able to secrete only early substrates), all expressing 3xFlag tagged OrgC, were analyzed by immunoblot with antibodies directed to the FLAG tag or the protein translocase SipB (as a control).
DOI: https://doi.org/10.7554/eLife.35886.002

The following figure supplement is available for figure 1:

**Figure supplement 1.** Genetic organization of different T3SS loci encoding OrgC homologs.
DOI: https://doi.org/10.7554/eLife.35886.003

To more directly test the potential involvement of OrgC in needle assembly we used an assay we previously developed to monitor needle length (*Lefebre and Galán, 2014*). We have previously observed that the absence of InvJ leads to the production of extremely long needles, which in turn result in the clumping of the bacterial culture due to needle tangling (*Lefebre and Galán, 2014*; *Kubori et al., 2000*). We therefore investigated the role of OrgC in needle formation by introducing a ΔorgC deletion in a ΔinvJ *S.* Typhimurium mutant strain, and examining its effect on needle-dependent bacterial clumping. We found that the deletion of *orgC* completely abolished the ΔinvJ *S.* Typhimurium cell clumping (*Figure 2C*) even though it did not alter the levels of the core components of the T3SS NC (*Figure 2D*). Consistent with this hypothesis and in sharp contrast with the ΔinvJ mutant, very few longer needles were observed on the surface of the *S.* Typhimurium ΔorgC ΔinvJ mutant when examined by electron microscopy (*Figure 2E*). Furthermore, we found that while PrgI secretion was increased in the ΔorgC mutant strain, introduction of a ΔinvJ mutation into this mutant background reduced the levels of secretion (*Figure 2—figure supplement 1*), presumably because in the absence of InvJ more PrgI subunits assemble into long needles. These results

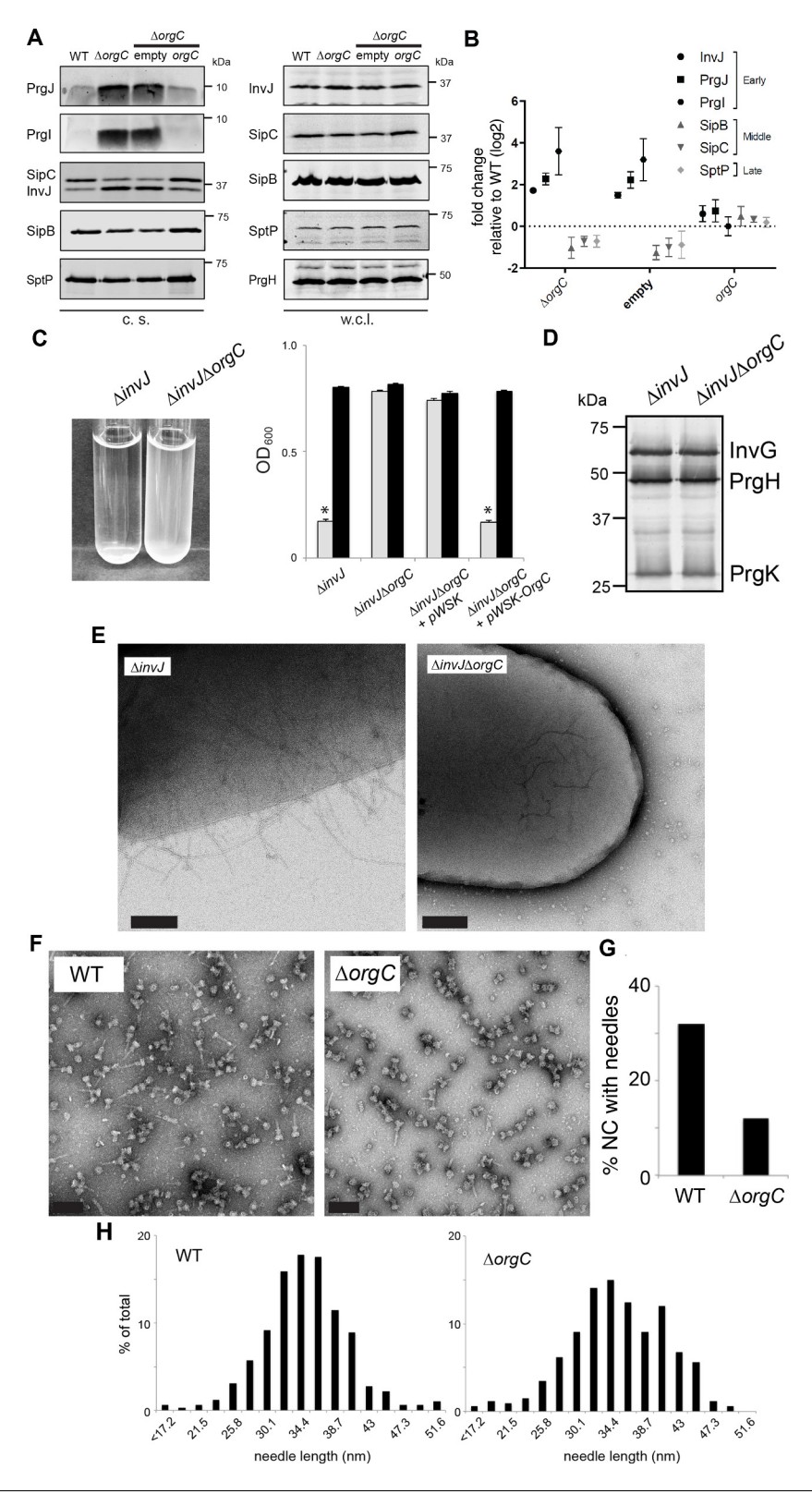

**Figure 2.** OrgC is necessary for efficient needle assembly but does not affect needle length. (**A**) *S.* Typhimurium Δ*orgC* secretes elevated amount of early and reduced amount of middle/late substrates. Proteins in bacterial culture supernatants (c. s.) were concentrated by TCA precipitation and analyzed by SDS-PAGE, followed by immunoblot using specific antibodies for PrgI, PrgJ, InvJ (early substrates), SipB, SipC (middle substrates) and SptP (late substrate). The secretion phenotype of Δ*orgC* was complemented by introducing a plasmid expressing OrgC but not by introducing the empty
*Figure 2 continued on next page*

*Figure 2 continued*

plasmid vector alone. Expression levels of the indicated proteins in whole-cell lysates (w. c. l.) were also evaluated. (B) Western-blot band intensities from three independent experiments were quantified. Values were normalized to the wild-type parent strain and log2 transformed. The dotted horizontal line corresponds to the levels of the different proteins in wild type *S.* Typhimurium. (C) A bacterial cell-clumping assay implicates OrgC in needle assembly. Cultures of the *S.* Typhimurium Δ*invJ* mutant strain, which display long needle filaments, clump and precipitate to the bottom of the tube (left panel) resulting in drastically decreased $OD_{600}$ (right panel, grey bars), which can be recovered by vortexing the samples (right panel, black bars). Bacterial cell clumping is abolished by introduction of a Δ*orgC* mutation (left and right panels), which can be complemented by introducing a plasmid encoding *orgC* (pWSK-*orgC*) but not by introducing the vector alone (pWSK) (right panel). Values represent $OD_{600}$ before (grey bars) and after (black bars) vortexing and are the mean ± SEM (standard error of the mean) of three independent measurements. Asterisks indicate statistically significant differences from the values of the vortexed sample (p<0.001, Student t test). (D) Western blot analysis of the NC base components InvG, PrgH, and PrgK in whole cell lysates of the Δ*invJ* and Δ*invJ* Δ*orgC S.* Typhimurium mutant strains. (E) Electron micrographs of negatively stained *S.* Typhimurium showing the presence (Δ*invJ*) or absence (Δ*invJ* Δ*orgC*) of long T3SS needle filaments. Scale bar = 200 nm, (Δ*invJ*); 100 nm (Δ*invJ* Δ*orgC*). (F) Electron micrographs of negatively stained needle complexes isolated from wild type (WT) or Δ*orgC S.* Typhimurium. Scale bar = 100 nm. (G) Percentage of needle complexes exhibiting the needle filament in preparations obtained from wild type (WT) or Δ*orgC S.* Typhimurium (number of particles analyzed: w. t. = 1105; Δ*orgC* = 1273). (H) Needle length of needle complexes isolated from *S.* Typhimurium wild type or Δ*orgC* mutant strains. The percentage of needle complexes exhibiting the indicated length (in nm, x axis) is indicated (number of needle complexes analyzed: w. t. = 314; Δ*orgC* = 339.

DOI: https://doi.org/10.7554/eLife.35886.004

The following figure supplement is available for figure 2:

**Figure supplement 1.** PrgI secretion profile of *S.*

DOI: https://doi.org/10.7554/eLife.35886.005

suggested that OrgC might be involved in needle formation. To further explore this hypothesis we isolated NCs from wild type and the Δ*orgC S.* Typhimurium mutants and compared them by electron microscopy. We found that the proportion of fully assembled NCs (i. e. NC bases with needle sub-structures) vs. NCs without needle substructures was significantly reduced in the *S.* Typhimurium Δ*orgC* mutant (*Figure 2F and G*). However, both the average length of the needles and the distribution of needle lengths in fully assembled NCs were indistinguishable in both strains (*Figure 2H*). Taken together, these results indicate that OrgC plays a role in the initiation of needle assembly but does not influence needle elongation or needle length.

## OrgC can exert its function when exogenously applied to bacterial cells

It is well established that type III secretion machines recognize substrates through signals located at their amino termini (*Michiels and Cornelis, 1991*; *Sory and Cornelis, 1994*; *Rüssmann et al., 2002*). Therefore to investigate whether the function of OrgC requires its secretion through the T3SS we constructed a mutant that lacks its first 21 amino acids. In addition, we constructed a mutant in which a maltose-binding protein (MBP) tag was placed at the amino terminus of OrgC, an arrangement that interferes with type III protein secretion (*Kubori and Galán, 2002*). Both mutant constructs were stably expressed in a *S.* Typhimurium Δ*orgC* mutant strain but, in contrast to wild type OrgC, they were not detected in the culture supernatants indicating that their secretion was abolished (*Figure 3A and B*). The ability of both mutant constructs to complement a Δ*orgC* mutant strain was then tested by examining their ability to complement the OrgC-dependent bacterial cell clumping phenotype observed in the Δ*invJ S.* Typhimurium mutant strain. We found that although introduction of a plasmid encoding wild type OrgC was able to complement the clumping phenotype of the *S.* Typhimurium Δ*orgC* Δ*invJ* mutant, plasmids encoding either of the non-secreted forms of OrgC did not (*Figure 3C*). These results indicate that OrgC secretion is necessary for its function.

The observation that OrgC must be secreted to exert its role in needle assembly suggested that it might be able to function from the outside of the bacterium. To test this possibility, we investigated whether exogenous addition of purified OrgC could complement the clumping phenotype of a Δ*orgC* Δ*invJ S.* Typhimurium mutant. We found that addition of purified OrgC to cultures of the *S.* Typhimurium Δ*invJ* Δ*orgC* strain resulted in bacterial cell clumping (*Figure 4A and B*) indicating that OrgC is capable of complementing the Δ*orgC* mutant when exogenously applied. The bacterial cell clumping observed was dependent on the presence of the Δ*invJ* mutation since exogenous addition of the OrgC protein did not induce clumping in the wild type strain or in a Δ*orgC* mutant alone (*Figure 4C*). Furthermore, the clumping was also strictly dependent on the presence of the needle protein PrgI and a functional T3SS since addition of the OrgC protein did not induce clumping in a

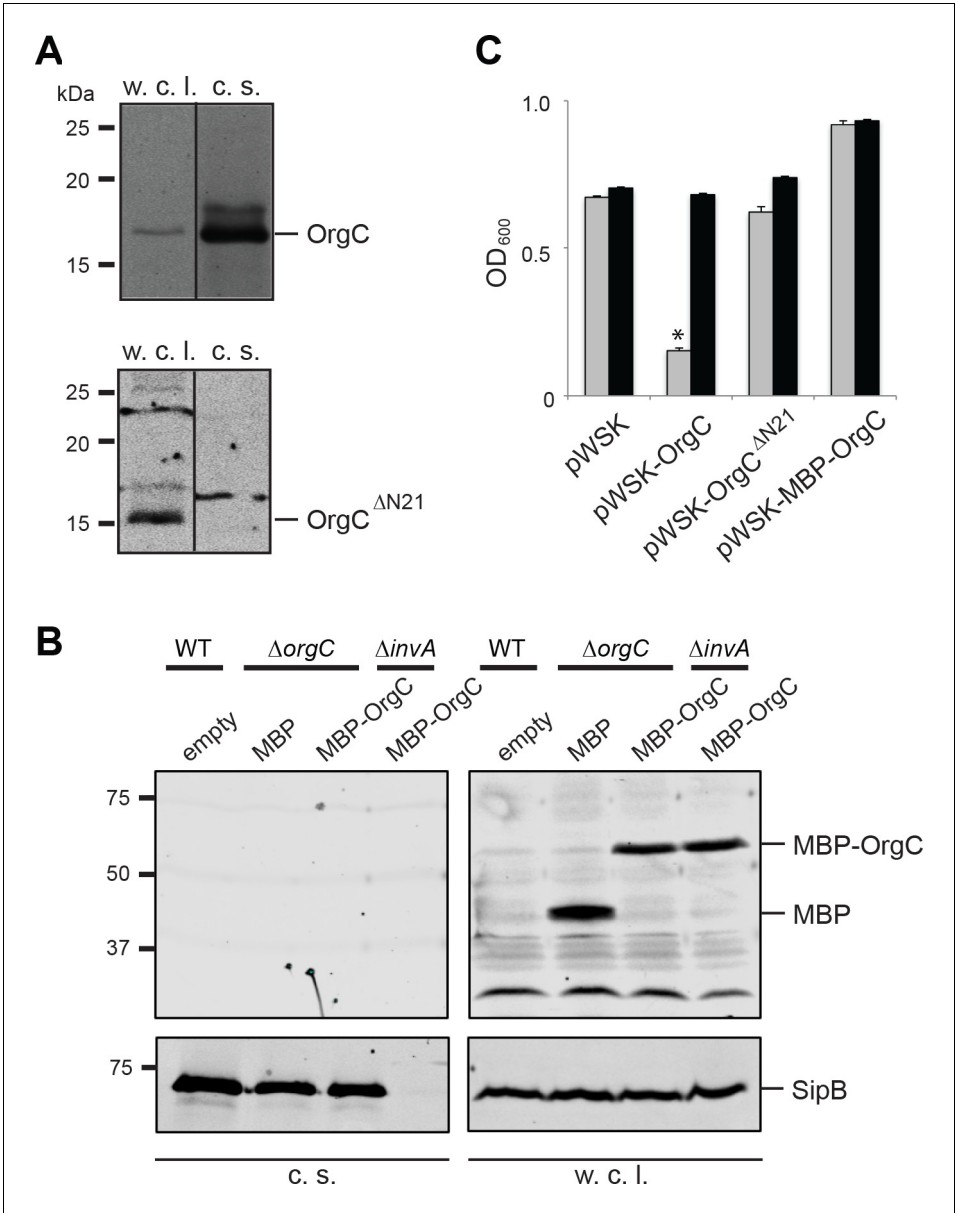

**Figure 3.** T3SS-mediated secretion of OrgC is required for its function. (A and B) Removal of its first 21 amino acids (A) or N-terminal addition of MBP (B) prevents the secretion of OrgC. Whole cell lysates (w. c. l.) and culture supernatants (c. s.) of wild-type *S.* Typhimurium expressing, C-terminally FLAG-tagged full length OrgC or an equivalently tagged deletion mutant lacking its first 21 amino acids (N21) were analyzed by western immunoblot with a monoclonal antibody directed to the FLAG tag (A). Alternatively, whole cell lysates (w. c. l.) or culture supernatants of wild-type *S.* Typhimurium, or the indicated isogenic mutants carrying an empty pWSK129 plasmid or its derivatives expressing either maltose-binding protein (MBP) or MBP-OrgC fusion were analyzed by western-blot using specific antibodies directed to the MBP tag or the protein translocase SipB (as a secretion control) (B). (C) Non-secretable forms of OrgC are non-functional. The *S.* Typhimurium Δ*invJ* Δ*orgC* mutant strains carrying the indicated plasmids were analyzed by the clumping assay as indicated in *Figure 2*. Values represent $OD_{600}$ before (grey bars) and after (black bars) vortexing and are the mean ± SEM of three independent measurements. Asterisks indicate statistically significant differences from the values of the vortexed sample (*p<0.001, Student t test).

DOI: https://doi.org/10.7554/eLife.35886.006

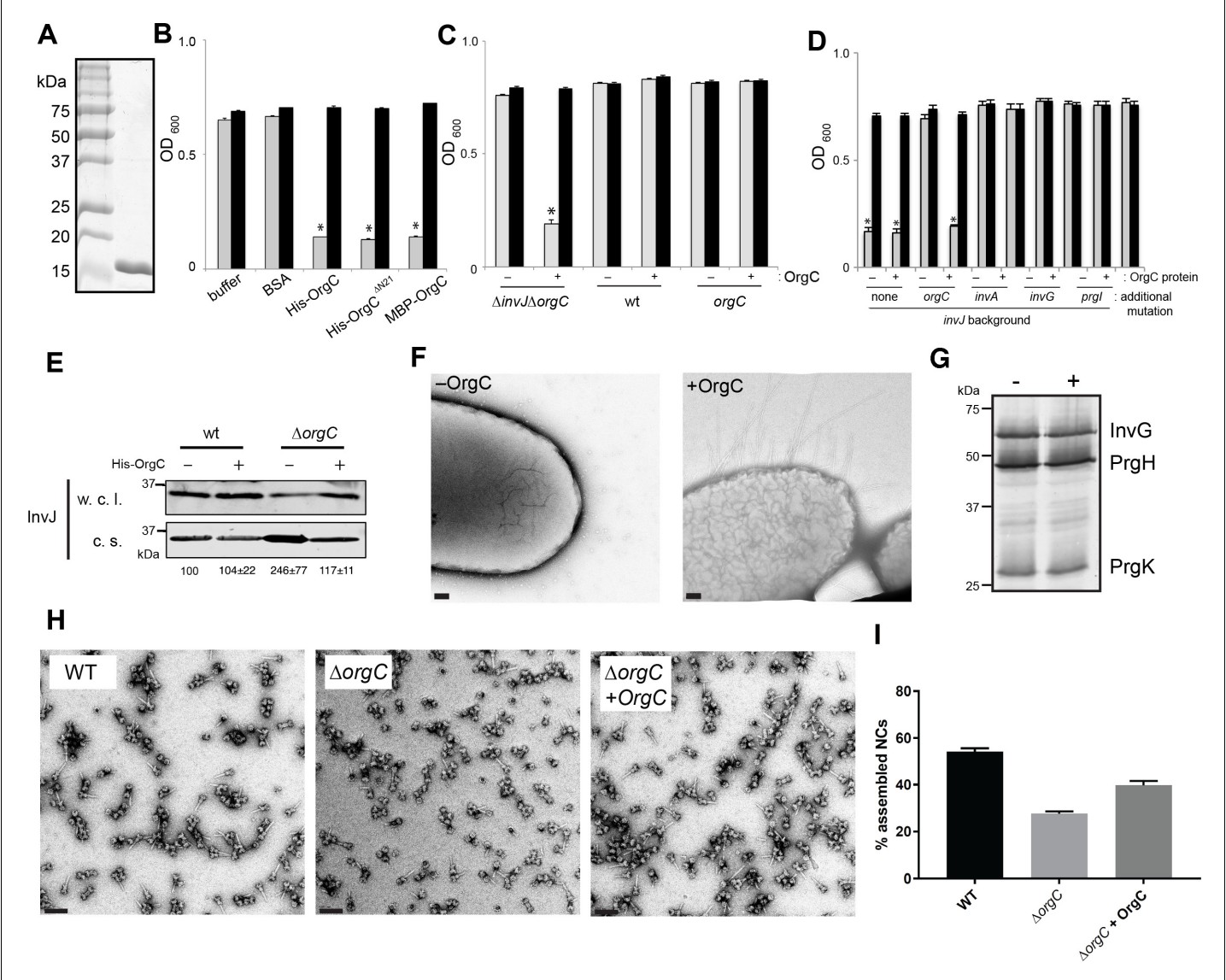

**Figure 4.** OrgC can exert its function when exogenously applied to bacterial cells. (A) His-OrgC was purified by Ni-affinity, ion exchange, and gel-filtration chromatography analyzed by SDS-PAGE and coomassie blue staining. (B – D) Administration of purified OrgC to bacterial cell culture can complement a ΔorgC mutation. *S.* Typhimurium ΔorgC ΔinvJ (B) or the indicated S. Typhimurium strains (C and D) were grown in the presence of the indicated protein preparations and the clumping of the bacterial cells was measured as indicated in *Figure 2*. Values represent $OD_{600}$ before (grey bars) and after (black bars) vortexing and are the mean ± SEM of three independent measurements. Asterisks indicate statistically significant differences from the values of the vortexed sample [*p<0.001 (B), p<0.005 (C and D), Student t test]. (E) The secretion profile abnormality of the ΔorgC mutant can be reversed by addition of purified OrgC. Secretion profile of ΔorgC *S.* Typhimurium grown in the presence or absence of purified OrgC. Proteins in the bacterial culture supernatant were concentrated by TCA precipitation and analyzed by Western blotting using specific antibodies directed to the early T3SS substrate InvJ. Numbers below the different lanes are the average ± SEM of the western-blot band intensities of InvJ in culture supernatants relative to wild-type (-)from three independent secretion assays. (F) Electron micrographs of negatively stained *S.* Typhimurium ΔorgC ΔinvJ grown in the presence (+OrgC) or in the absence (−OrgC) of purified OrgC. Note the presence of long filaments when bacteria are grown in the presence of purified OrgC protein. Scale bar: 100 nm. (G) Western blot analysis of the NC base components InvG, PrgH, and PrgK in whole cell lysates of *S.* Typhimurium ΔinvJ ΔorgC grown in the presence or absence of purified OrgC. (H) Electron micrographs of negatively stained needle complexes isolated from wild-type (WT), and ΔorgC *S.* Typhimurium mutant strains grown in the absence (−OrgC) or in the presence (+OrgC) of purified OrgC protein. Scale bar: 100 nm. (I) The proportion of needle complexes displaying the needle filament in the indicated strains was determined. Values are expressed as the mean percentage (±SEM) of needle complexes per micrograph exhibiting the needle filament. Number of particles analyzed from 28 micrographs: w. t. = 2979; ΔorgC (−OrgC) = 3334; ΔorgC (+OrgC) = 3957.

DOI: https://doi.org/10.7554/eLife.35886.007

$\Delta invJ/\Delta orgC/\Delta prgI$, $\Delta invJ/\Delta orgC/\Delta invA$, or $\Delta invJ/\Delta orgC/\Delta invG$ mutant strains (**Figure 4D**), which lack the needle substructure and/or are defective for type III secretion (**Sukhan et al., 2001**; **Collazo and Galán, 1996**; **Galán et al., 1992**; **Kaniga et al., 1994**). We also found that growth of the *S. typhimurium* $\Delta orgC$ mutant in the presence of purified OrgC resulted in decreased levels of the early substrate protein InvJ in the cultured supernatants (**Figure 4E**), which is consistent with the observed presence of a larger number of fully assembled NC in the presence of OrgC and presumably a more efficient substrate switching (see **Figure 2E and F**). We then examined by electron microscopy bacterial cells from the *S.* Typhimurium $\Delta invJ \Delta orgC$ mutant grown in the presence or absence of purified OrgC. We found the presence of abundant long needle structures protruding from the surface of the mutant strain when grown in the presence of OrgC but not when grown in its absence (**Figure 4F**), despite the presence of equal levels of the components of the NC after growth under either condition (**Figure 4G**). We also isolated needle complexes from the *S.* Typhimurium $\Delta orgC$ mutant grown in the presence or absence of purified OrgC. We found that the proportion of fully assembled NCs (i. e. NC bases with needle substructures) vs. NCs without needle substructures was increased when the $\Delta orgC$ mutant was grown in the presence of the purified OrgC protein (**Figure 4H and I**). Taken together, these results indicate that OrgC can stimulate needle assembly when exogenously applied to culture cells and suggest that OrgC must exert its function at a location within the needle complex that is accessible from the outside.

## OrgC interacts with PrgI and accelerates needle polymerization in vitro

The observation indicating that OrgC contributes to efficient needle filament assembly prompted us to investigate its potential interaction with the needle subunit PrgI in *S.* Typhimurium cells. Using an affinity purification assay with a tagged version of OrgC we detected the interaction of OrgC with PrgI but not with the other type III secretion early substrates PrgJ and InvJ (**Figure 5A**). To investigate whether the interaction of OrgC with PrgI was direct, we used a bacterial two-hybrid system in a heterologous host (*E. coli*). Using this approach we readily detected the interaction between OrgC and PrgI$^{\Delta C5}$ (**Figure 5B**), a mutant that lacks its last five residues and consequently is unable to polymerize thus preventing protein agregation (**Poyraz et al., 2010**). These results indicate that the interaction between OrgC and PrgI is direct. We hypothesized that if this direct interaction enhances the efficiency of needle filament assembly, overexpression of PrgI might suppress the $\Delta orgC$ phenotype. Consistent with this hypothesis, when *prgI* was over-expressed in a $\Delta invJ \Delta orgC$ background, the cell clumping was recovered despite the absence of OrgC (**Figure 5C**). These results indicate that the mechanism by which OrgC promotes needle assembly involves its direct interaction with the needle filament protein itself.

It is well established that purified T3SS needle proteins can self-polymerize in vitro and that polymerization can be monitored by dynamic light scattering (DLS) (**Poyraz et al., 2010**). We therefore examined the effect of adding purified OrgC to a PrgI needle protein polymerization reaction in vitro using DLS. We found that OrgC significantly accelerated the polymerization of PrgI into needle filaments (**Figure 5D**). Under the assay conditions used, recombinant PrgI polymerized into needles at about 120 min after initiation of the reaction. In contrast, in the presence of recombinant OrgC, PrgI polymerization could be detected as early as 30 min after initiation of the reaction (**Figure 5D**). Taken together, these results indicate that OrgC directly facilitates the assembly of the PrgI needle filaments.

## The interaction of OrgC with PrgI is required for its function

To further characterize the interaction between OrgC and PrgI we used NMR spectroscopy. Titrations of OrgC with PrgI showed direct and specific protein-protein interaction between OrgC and PrgI (**Figure 6A–D** and **Figure 6—figure supplement 1**). The primary sites of interaction involved the OrgC residues 134–141 at the C-terminal region, and the PrgI residues K37, S39, Q48, and K66 at the head group of the alpha-helical hairpin and the C-terminal helix of PrgI (**Figure 6A–6E**). These residues showed the largest changes in the NMR peak positions of the backbone amides during the titration of PrgI (**Figure 6A and C** and **Figure 6—figure supplement 1**). The change in peak positions during the titration indicated that the interaction occurred at fast exchange NMR time scale, which is suggestive of weak binding interaction typically seen for binding interactions with micromolar binding affinities. Interestingly, amino acid sequence comparison of needle filament proteins

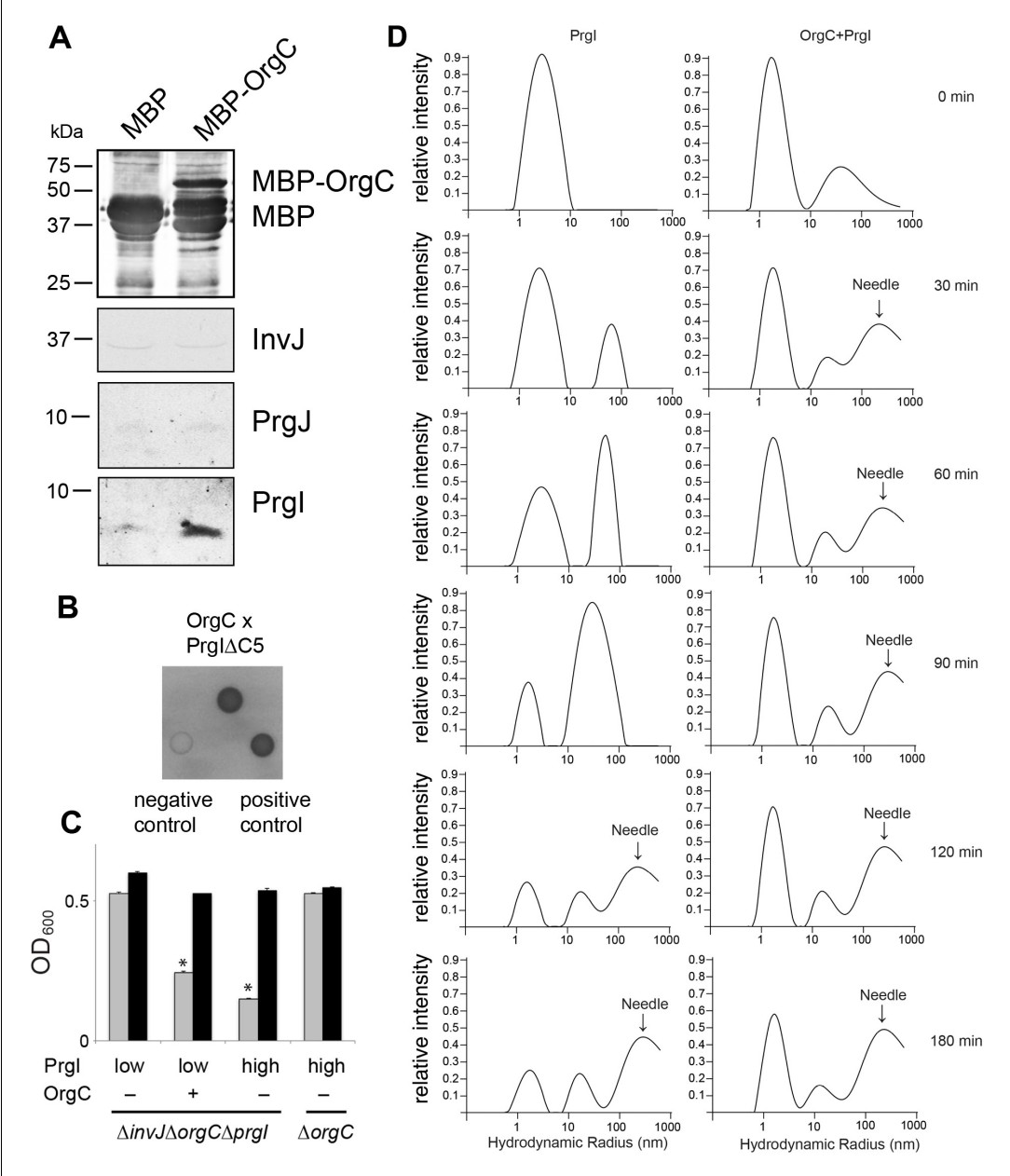

**Figure 5.** OrgC interacts with the needle filament protein PrgI and accelerates its in vitro polymerization. (A) MBP-tagged OrgC or MBP alone were expressed in a Δ*orgC S.* Typhimurium strain and affinity-purified from whole cell lysates with amylose resin. Bound proteins were eluted with maltose and analyzed by Western blotting with the indicated antibodies. A specific PrgI signal was detected in the MBP-OrgC sample but not in the MBP control. Interaction of MBP-OrgC with other early T3SS substrates (PrgJ or InvJ) was not detected. (B) The interaction of OrgC with PrgI was examined in a bacterial adenylate cyclase-reconstitution two-hybrid assay. *E. coli* strains carrying plasmids encoding bait and target proteins fused to a PrgI deletion mutant lacking its five last residues (PrgIΔC5), and OrgC, respectively. Plasmids encoding the leucine zipper domain of the yeast transcription activator GCN4 (Zip) or strains carrying empty vectors served as positive and negative controls, respectively. All strains were plated on MacConkey agar to visualize interactions. Colonies of bacteria expressing interacting proteins appear green (dark in this figure). (C) Effect of PrgI overexpression on the Δ*orgC* phenotype. PrgI was expressed in a Δ*invJ* Δ*orgC* Δ*prgI* or Δ*orgC S.* Typhimurium strains from an arabinose-inducible promoter after growth in the presence of low (0.00075%) or high (0.015%) levels of arabinose, and bacterial cell clumping was assayed as indicated in *Figure 2*. Values represent OD$_{600}$ before (grey bars) and after (black bars) vortexing and are the mean ± SEM of three independent measurements. Asterisks indicate statistically significant differences from the values of the vortexed sample (*p<0.001; Student *t* test). (D) OrgC stimulates PrgI polymerization in vitro. The effect of OrgC on the polymerization of recombinant PrgI into needles was monitored by dynamic light scattering (DLS) from 0 to 180 min. In the absence of OrgC, PrgI polymerized into needles at ~120 min. Addition of OrgC at 1:1 molar ratio accelerated the polymerization of PrgI into needles, which could be detected after 30 min. DLS peaks were assigned as monomeric PrgI (peak below 10 nm hydrodynamic radius), PrgI oligomers (10–100 nm), and PrgI

*Figure 5 continued on next page*

*Figure 5 continued*

polymers or needles (100–1000 nm). As control, OrgC did not show the needle DLS peak between 100–1000 nm and remained well below 100 nm hydrodynamic radius.

DOI: https://doi.org/10.7554/eLife.35886.008

showed a higher degree of conservation of amino acids involved in OrgC binding in the T3SSs that have homologs of OrgC relative to needle filament proteins from those that do not (*Figure 6—figure supplement 2*). Taken together, these results indicate that OrgC directly interact with PrgI through a discrete domain at its carboxy terminus and identified critical PrgI residues involved in this interaction.

To determine if the interaction between OrgC with PrgI is required for OrgC function, we mutated the residues in both proteins mapped by the NMR titration experiments and examined the resulting mutants in functional assays for needle filament assembly. Introduction of small deletions (Δ134–136 and Δ137–140) within the predicted PrgI-binding site of OrgC abolished its ability to complement bacterial cell clumping of a Δ*orgC* Δ*invJ* mutant strain (*Figure 6F*), even though introduction of the mutations did not affect the stability of OrgC (*Figure 6—figure supplement 3*). Similarly, introduction of mutations in the PrgI residues that showed significant changes in the NMR peak positions (Q48, S52 and N55) during the titration of OrgC reduced PrgI-mediated bacterial cell clumping (*Figure 6F*). The inability of the PrgI mutants to interact with OrgC would be expected to result in increased secretion of early substrates and decrease secretion of middle and late substrates. However, we found that introduction of the multiple mutations in PrgI did not significantly alter type III secretion (*Figure 6—figure supplement 4*). We hypothesize that the rather complex interface between OrgC and PrgI may be difficult to disrupt by discrete mutagenesis without globally altering needle polymerization. Consequently, the disruption of the OrgC/PrgI interface by introducing mutations in selected amino acids, while sufficient to result in a phenotype measurable by the more sensitive bacterial cell-clumping assay, it was insufficient to result in a phenotype that can be observed with the less sensitive secretion assay. In any case, these results indicate that the introduction of the PrgI mutations did not alter needle assembly. Taken together, these results validate the NMR titration experiments and demonstrate that the interaction of OrgC with PrgI is essential for its function.

## OrgC forms a 4-helix structure with a novel fold

To further understand the structural basis for the function of OrgC we sought to determine its atomic structure. Attempts at crystallization of OrgC did not yield crystals suitable for high-resolution X-ray structure determination. Therefore, we determined the 3D structure of OrgC by NMR methods. Full-length recombinant OrgC expressed poorly, showed poor solubility in NMR solution conditions, and the quality of NMR spectra showed poor resolution and heterogeneity in peak intensities suggestive of non-uniform protein conformations (*Figure 7—figure supplement 1*). Consequently, to facilitate structural determination we removed the first 20 amino acids from OrgC, which harbors its secretion signal and therefore is predicted to be disordered. The truncated form of OrgC showed excellent quality of NMR spectra (*Figure 7—figure supplement 1*) with well-resolved peaks of homogenous intensities that enabled the NMR structure determination. NMR peak assignments were obtained from 2D $^{15}$N HSQC (*Grzesiek and Bax, 1993a*) and 3D HNCACB (*Wittekind and Mueller, 1993*), HNCA (*Grzesiek et al., 1992*), CACBCONH (*Grzesiek et al., 1992*), and HBHA(CO) NH (*Grzesiek and Bax, 1993b*). Analysis of the $^{13}$Cα, $^{13}$Cβ and $^{1}$Hα secondary chemical shifts (*Wishart and Nip, 1998*) identified the helical regions of OrgC (*Figure 7—figure supplement 2*). For structure calculations, distance restraints were obtained from nuclear Overhauser effects (NOEs) assigned from 3D $^{15}$N-edited HSQC NOESY ($t_{mix}$120 ms) and 3D $^{13}$C-edited HMQC NOESY ($t_{mix}$120 ms). Using intra-residue NOEs as internal standards, the NOEs were categorized based on peak intensities into strong, medium, and weak NOEs, and assigned upper distance limits of 2.7 Å, 3.5 Å, and 5.5 Å, respectively. Dihedral angle restraints of phi −60 ± 20° and psi −40 ± 20° were used for alpha helical regions identified by the secondary chemical shifts. The iterative calculation of multiple structures by simulated annealing and molecular dynamics followed by energy minimization yielded a high-confidence structural model for OrgC. The structure was derived from 1170 non-redundant

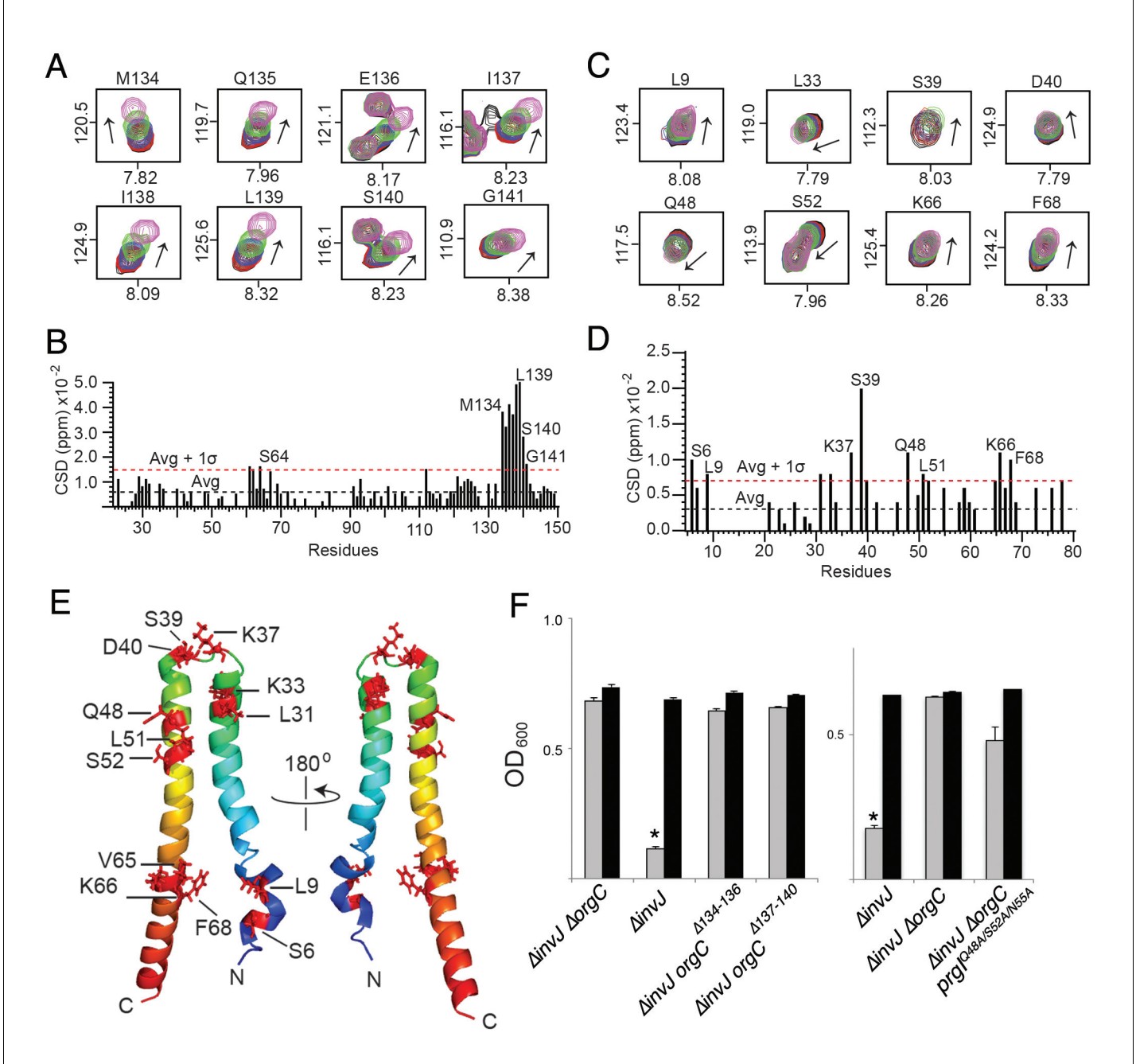

**Figure 6.** NMR titrations of OrgC and PrgI interaction. (A) Selected $^{1}$H-$^{15}$N peaks of $^{15}$N labeled OrgC that showed the largest changes upon titration with unlabeled PrgI. (B) Plot of the weighted chemical shift deviation of $^{15}$N OrgC titrated with PrgI. (C) Selected $^{1}$H-$^{15}$N peaks of $^{15}$N labeled PrgI that showed the largest changes upon titration with unlabeled OrgC. (D) Plot of the weighted chemical shift deviation of $^{15}$N PrgI titrated with OrgC. In (A) and (C), peaks are colored according to the molar ratios of $^{15}$N protein: unlabeled binding partner as follows: black (1:0), red (1:0.5), blue (1:1), green (1:2), and pink (1:4). In (B) and (D), dashed lines show the average and average plus one standard deviation (σ) of the chemical shift deviation. (E) Location in the atomic structure of PrgI residues that showed the largest changes in the NMR peak positions of the backbone amides upon titration with OrgC. (F) Effect of mutations in the interacting residues in OrgC and PrgI. The indicated *S.* Typhimurium mutant strains were analyzed by the clumping assay as indicated in *Figure 2*. Values represent $OD_{600}$ before (grey bars) and after (black bars) vortexing and are the mean ± SEM of three independent measurements. Asterisks indicate statistically significant differences from the values of the vortexed sample (*p<0.001, Student *t* test).
DOI: https://doi.org/10.7554/eLife.35886.009

The following figure supplements are available for figure 6:

**Figure supplement 1.** NMR titrations of OrgC and PrgI.

*Figure 6 continued on next page*

*Figure 6 continued*

DOI: https://doi.org/10.7554/eLife.35886.010

**Figure supplement 2.** PrgI residues (Ser39 and Ser52) directly involved in OrgC binding are exclusively conserved in PrgI homologs that have an OrgC partner.

DOI: https://doi.org/10.7554/eLife.35886.011

**Figure supplement 3.** Stability of the OrgC protein deletion mutants.

DOI: https://doi.org/10.7554/eLife.35886.012

**Figure supplement 4.** Mutations in the PrgI residues important for its interaction with OrgC do not affect type III secretion.

DOI: https://doi.org/10.7554/eLife.35886.013

NOEs, with an average of about nine restraints per residue, and the packing of the hydrophobic core of the protein was defined by 173 long range NOEs. The 20 low energy NMR structures showed good convergence into a single family of structures with low violations in distance and dihedral angle restraints (*Figure 7—figure supplement 3*). Finally, over 99% of the residues are in the allowed phi and psi regions of the Ramachandran plot (*Supplementary file 1*).

The overall structure of OrgC comprises a central region with four helices – helix α1 (residues 46–57), helix α2 (residues 55–78), helix α3 (residues 92–105), and helix α4 (residues 111–126) (*Figure 7A*) – bounded by partially disordered regions at its amino and carboxy terminal boundaries. The four helices pack into a volume that resembles a triangular prism with a length shorter than the base or height of the two triangular faces of this prism. Helix α1 and α2 form one face that defines the base and height of this triangular prism. Helix α3 and α4 form the second triangular face, and the packing of helix α1/α2 to helix α3/α4 defines the short length of this triangular prism-shaped domain. The N-terminal region from residues 25 to 36 has partial helical character based on the small Cα secondary chemical shifts (*Figure 7—figure supplement 3*). This partially folded region is connected by a loop (residues 37–45) to the core 4-helix domain of OrgC.

The PrgI-binding region of OrgC (residues 134–141) falls within an 8-residue flexible loop that is dangling from the 4-helix central core domain of OrgC (*Figure 7A*). Within this loop, residues 133–139 have a somewhat partial helical character as indicated by the small Cα secondary chemical shifts (*Figure 7—figure supplement 2*). Nevertheless, this PrgI-binding region is essentially lacking in tertiary structure. Further analysis of the electrostatic surface of OrgC indicated that the PrgI-binding region is mostly electronegative (*Figure 7B*). The corresponding surface in PrgI involved in OrgC binding is mostly electropositive (*Figure 7C*), suggesting a role for electrostatic interactions in the binding mechanisms.

To get more insight into OrgC function we generated a model of how it might dock in the assembled needle filament (*Figure 7D and E*). The NMR titration experiments identified two surfaces in PrgI that are involved in its interaction with OrgC (*Figure 6E*): a surface involving residues S39 and L51 located towards the head group of the PrgI alpha helical hairpin, and another surface involving residue F68 located towards the bottom of this hairpin (*Figure 6E*). In the model, the PrgI surface defined by its residues S39 and L51 is exposed at the tip of the assembled needle filament, and is therefore accessible for OrgC binding, specifically to its residues 134–141 (*Figure 7D*). The second PrgI surface defined by F68, however, is primarily involved in PrgI-PrgI contacts in the assembled needle and therefore would not be available for OrgC binding in the assembled needle filament. We propose a model in which OrgC would bind PrgI prior to its assembly into the needle filament engaging both interacting surfaces, the one defined by S39/L51 and that defined by F68. Such interaction would presumably lead to a more efficient needle assembly. Once PrgI polymerized into the nascent filament, OrgC would disengage from PrgI as the F68 surface would be now engaged in high-affinity PrgI-PrgI contacts. A corollary of this model is that OrgC would not be able to bind the tip of the assembled needle filament due to the absence of a critical PrgI surface defined by PrgI[F68]. This prediction is consistent with our inability to detect OrgC at the tip of the growing needle filament.

A database search for related structures using DALI (*Holm and Sander, 1993*) detected no structural homologs, thus suggesting that OrgC represent a unique fold. Of interest is the observation that no true structural homologs could be detected among the components of the flagellar apparatus, which is thought to be the evolutionary precursor of virulence-associated type III secretion systems. However, it is noteworthy that the D1 domain of the flagellar capping protein FliD loosely

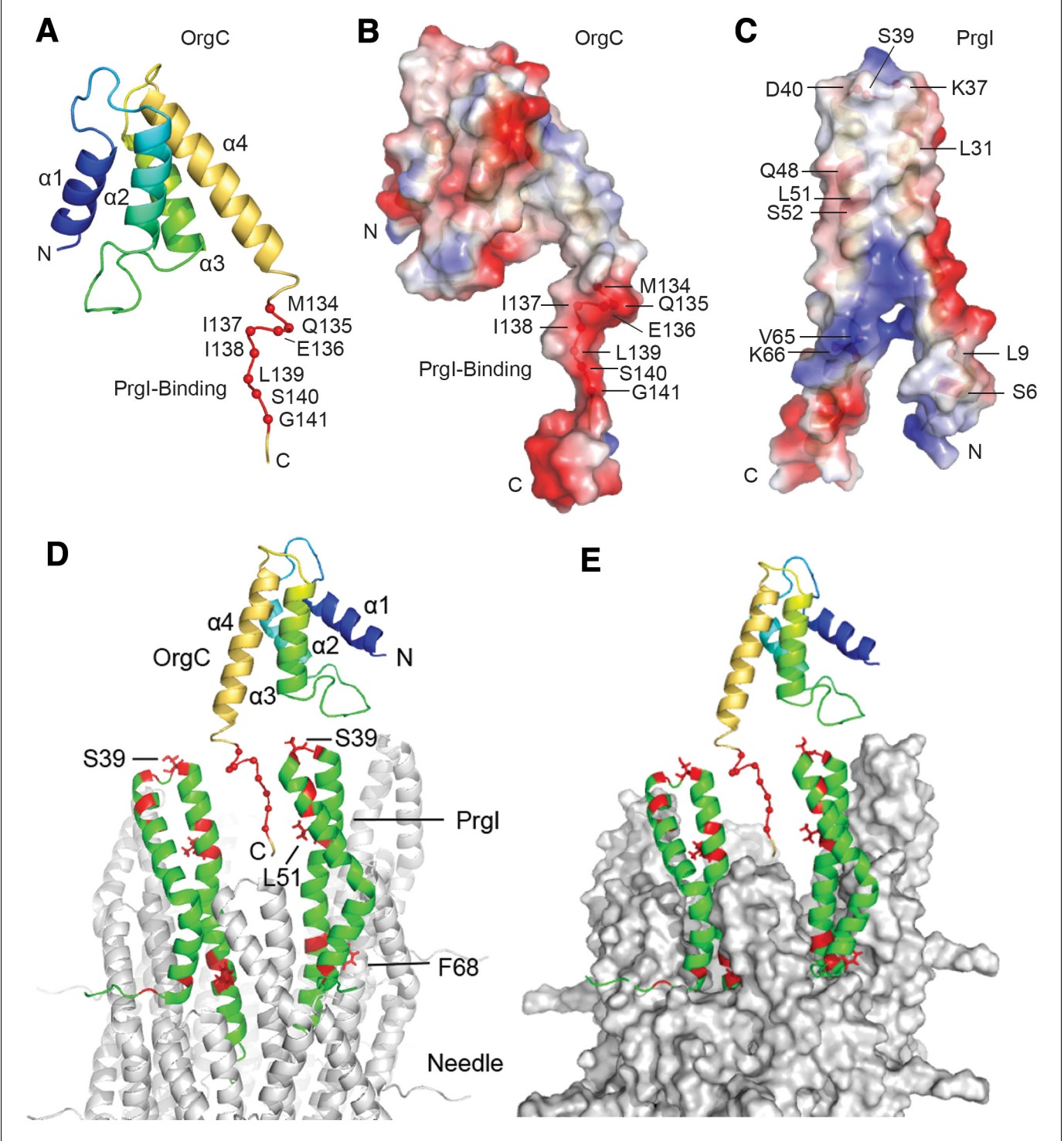

**Figure 7.** NMR structure of OrgC and structural model for its binding to PrgI in the assembled needle filament. (**A and B**) The structured domain of OrgC consists of a 4-helix bundle and the PrgI-binding region is located at the C-terminus of the OrgC domain (**A**). The surface of the PrgI-binding region of OrgC is electronegative (**B**), which complements the electropositive surface on its binding partner, PrgI (**C**). (**D and E**). A model of OrgC with the assembled PrgI needle filament. The tip of the assembled PrgI needle is shown as a ribbon (**D**), and as a surface representation (**E**). The structure of the needle filament was from Loquet et al (**Loquet et al., 2012**). Two adjacent PrgI monomers at the needle tip are shown in green ribbon and the residues identified to be involved in binding OrgC are shown in red. There are spaces between the PrgI monomers at the needle tip that expose some of the PrgI residues (i.e. S39 and L51) involved in the interaction with OrgC. Residues near F68 are involved in PrgI-PrgI contacts in the assembled needle.

*Figure 7 continued on next page*

*Figure 7 continued*

DOI: https://doi.org/10.7554/eLife.35886.014

The following figure supplements are available for figure 7:

**Figure supplement 1.** Comparison between the 2D proton-nitrogen correlation NMR spectra of (**A**) full length OrgC and (**B**) N-terminal truncation construct of OrgC.

DOI: https://doi.org/10.7554/eLife.35886.015

**Figure supplement 2.** Secondary (**A**) $^{13}C\alpha$, (**B**) $^{1}H\alpha$, and (**C**) $^{13}C\beta$ chemical shifts of OrgC identified four helices (α1, α2, α3, and α4) in the structure of OrgC.

DOI: https://doi.org/10.7554/eLife.35886.016

**Figure supplement 3.** Superposition of the lowest energy 20 NMR structures of OrgC, (**A**) showing only the structured 4-helix domain of OrgC from residues 44–130, and (**B**) shown with the flexible PrgI-binding region, residues 134–141 (red lines and spheres).

DOI: https://doi.org/10.7554/eLife.35886.017

**Figure supplement 4.** The topology of the OrgC fold shares similarity only partly with the topology of the FliD D1 domain.

DOI: https://doi.org/10.7554/eLife.35886.018

shares some of the topological features of the OrgC central four-helix domain (*Figure 7—figure supplement 4*). The functional implications of this observation, if any, are unknown but may suggest some convergent evolutionary process in the emergence of OrgC.

## Discussion

This study describes an as of yet undiscovered component of non-flagellar T3SSs, which is encoded within *S.* Typhimurium SPI-1, next to the cluster of genes that encode the T3SS injectisome. In *S.* Typhimurium this gene was originally identified as co-regulated with other T3SS-associated genes within the SPI-1 cluster, and more specifically co-regulated in response to oxygen levels (hence his name o̲xygen r̲egulated g̲ene C or OrgC) (*Jones and Falkow, 1994*; *Klein et al., 2000*). We found that OrgC is an 'early' substrate of the *S.* Typhimurium SPI-1 T3SS, co-secreted with the inner rod and needle filament proteins, and the regulatory protein InvJ. Consistent with its position in the secretion hierarchy, our results indicate that OrgC plays a role in the initiation of assembly of the needle filament of the injectisome. We arrive at this conclusion based on several pieces of evidence. First, we found that removal of *orgC* from a Δ*invJ S.* Typhimurium effectively abolished the clumping phenotype observed in this mutant strain due to the entanglement of the long needles that result from the absence of InvJ. Second, this effect was not associated with the presence of shorter needles but rather, with the presence of a reduced number of long needles on the bacterial surface of the Δ*orgC* Δ*invJ* double mutant. Consistent with these observations, deletion of *orgC* from wild type *S.* Typhimurium did not alter needle length but significantly reduced the number of NCs harboring an attached needle substructure. Third, secretion of middle and late substrates was also reduced in the Δ*orgC* mutant strain. This observation is in agreement with the reduction in the number of fully assembled injectisomes in this mutant strain, which is a requirement for the secretion of middle and late substrates by the T3SS. Finally, we found that OrgC directly interacts with PrgI and accelerates the initiation of its polymerization in vitro.

What is the mechanism by which OrgC may exert its function? Our results indicate that OrgC directly interacts with PrgI both in vivo and in vitro as shown by co-immunoprecipitation, bacterial two-hybrid and NMR spectroscopy assays. Mutations in residues in both OrgC and PrgI determined to be critical for binding by NMR spectroscopy resulted in a loss of OrgC function. We also found that OrgC must be secreted by the T3SS to exert its function and that a Δ*orgC* mutation can be complemented by the exogenous addition of purified OrgC protein. Importantly, dynamic light scattering experiments showed that purified OrgC was able to accelerate the in vitro polymerization of PrgI into needle filaments. Taken together these results indicate that to exert its function OrgC must physically interact with PrgI and that the place of this interaction must be accessible from the outside. Therefore, these observations suggest a mechanism by which OrgC may help the initiation of the polymerization of PrgI into the needle filament. Our results indicate that OrgC may not be necessary for the subsequent elongation of the needle substructure since needles assembled in the Δ*orgC* mutant, although significantly reduced in numbers, exhibited the same length as those assembled in strains expressing OrgC. If OrgC were required for needle elongation, needle

filaments would be expected to be shorter. Furthermore, we were not able to detect the presence of OrgC at the tip of growing needle filaments or in isolated needle complexes, which would also be expected if OrgC were involved in needle elongation. Therefore, we hypothesize that OrgC may help to nucleate the initial polymerization of PrgI monomers. Once assembly is initiated, OrgC may be discarded thus explaining our inability to detect it at the tip of the growing needle filaments or as an integral component of isolated NCs. Consistent with this hypothesis, the NMR titration experiments identified PrgI residues critical for OrgC binding that are not available for binding in the assembled filament further supporting this model.

The function of OrgC appears to be substantially different from another protein reported to influence needle assembly in *Yersinia* spp. that may involve the control of the secretion of the needle protein (*Blaylock et al., 2010*). However, the activity of OrgC and the phenotype of the Δ*orgC* mutant are reminiscent of flagellar capping proteins, which are required for the assembly of the rod, hook, and filament, the flagellar substructures that make up the entire flagellum (*Ikeda et al., 1993*; *Ohnishi et al., 1994*; *Yonekura et al., 2000*). Each of the different flagellar substructures has its own specific capping protein, which control their distal growth by regulating the assembly of the different subunits at the growing tip. The recent atomic and cryo EM structures of FliD, the capping protein for the flagellar filament, have provided insight into the potential mechanism by which these proteins may exert their function (*Maki-Yonekura et al., 2003*; *Postel et al., 2016*; *Song et al., 2017*). FliD consists of three domains (D1, D2, and D3) that assemble into an hexamer (in the case of *E. coli* FliD) or a pentamer (in the case of *S.* typhimurium FliD) (*Song et al., 2017*). In the multimeric forms, the D2 and D3 domains form a plate that resembles either a 5- or 6-point star. The D1 domain, which resembles flagellin, forms 5 or 6 leg-like structure, which connects the plate to the tip of the flagella. A model has been proposed in which the D1 domain of FliD temporarily occupies a position in place of a nascent flagellin until the flagellin reaches the growing end of the filament, at which point FliD would move aside to repeat the cycle (*Song et al., 2017*; *Yonekura et al., 2000*). In this model, the plate structure made of domains D2 and D3 would slow down the transit of flagellar subunits to facilitate their assembly at the tip. There are many similarities between OrgC and capping proteins. Like capping proteins, OrgC is secreted through the T3SS, promotes the assembly of the needle filament, and can exert its function when exogenously added to culture media. However, there are many significant differences as well. Unlike capping proteins, OrgC appears to be required only for the efficient initiation of needle filament assembly but not for its elongation. Once assembly is initiated, OrgC is likely discarded to the culture supernatant as we were unable to detect it bound to the tip of the growing needle or in isolated NCs. Furthermore, unlike the capping proteins, OrgC is not essential for needle assembly. In its absence, the number of fully assembled NC is reduced although those that assemble exhibit needle filaments that are of the same length as those of wild type. The functional differences between OrgC and the capping proteins are also reflected in substantial differences in their sizes (150 vs ~450 amino acids) and amino acid sequence. However, the NMR structure of OrgC indicates that even though it shares no amino acid sequence similarity with capping proteins, its central core domain exhibits topological similarities with the D1 domain of capping proteins (*Deane et al., 2006*; *Wang et al., 2007*; *Yonekura et al., 2000*). Therefore it is possible that OrgC may be equivalent to capping proteins lacking the 'plate' substructure, which may explain the substantial differences in the activities of these functionally related proteins. The diameter of the T3SS needle is significantly smaller than the diameter of the flagellar filament (8 nm vs 20 nm), which may translate into differences in its assembly mechanism, and in turn, differences in the requirement and/or function of the accessory proteins during assembly. In fact, OrgC homologs appear to be absent from several T3SSs, suggesting that the needle filaments of these systems may have evolved a complete independence from accessory proteins for their assembly. In support of this hypothesis, we detected a high degree of conservation in the amino acids involved in the interaction with OrgC in needle filament proteins from T3SS that have homologs of OrgC (*Figure 6—figure supplement 2*).

In summary, we have described an as of yet unidentified component of non-flagellar T3SSs, which promotes the assembly of the needle filament substructure. These studies provide major insight into the assembly mechanisms of the T3SS injectisome and highlight differences and similarities with the evolutionary related flagellar apparatus.

# Materials and methods

## Key resources table

| Reagent type (species) | Designation | Source or reference | Identifiers | Additional information |
|---|---|---|---|---|
| Strain, strain background (*Salmonella enterica* serovar Typhimurium) | SB300 | Nature, 291:238 | wild type | Mouse isolate of SL1344 |
| Strain, strain background strain background (*Salmonella enterica* serovar Typhimurium) | SB2942 | This study | *orgC 3xFlag invC::kan* | |
| Strain, strain background (*Salmonella enterica* serovar Typhimurium) | SB2943 | This study | *orgC 3xFlag invJ::kan* | |
| Strain, strain background (*Salmonella enterica* serovar Typhimurium) | SB2946 | This study | *orgC 3xFlag spaS-3xFlag* | |
| Strain, strain background (*Salmonella enterica* serovar Typhimurium) | SB2947 | This study | *orgC 3xFlag spaSN258A-3xFlag* | |
| Strain, strain background (*Salmonella enterica* serovar Typhimurium) | SB2326 | This study | Δ*invJ flhD::Tn10* | |
| Strain, strain background (*Salmonella enterica* serovar Typhimurium) | SB2939 | This study | Δ*invJ*Δ*orgC flhD::Tn10* | |
| Strain, strain background (*Salmonella enterica* serovar Typhimurium) | SB3079 | This study | *mbp-prgH flhD::tet* | |
| Strain, strain background (*Salmonella enterica* serovar Typhimurium) | SB3275 | This study | *mbp-prgH flhD::tet* Δ*orgC* | |
| Strain, strain background (*Salmonella enterica* serovar Typhimurium) | SB762 | Infect. Immun. 68:2335 | *flhD::Tn10* | |
| Strain, strain background (*Salmonella enterica* serovar Typhimurium) | SB1679 | This study | Δ*orgC* | |
| Strain, strain background (*Salmonella enterica* serovar Typhimurium) | SB2639 | This study | Δ*invJ flhD::Tn10* | |
| Strain, strain background (*Salmonella enterica* serovar Typhimurium) | SB2944 | This study | Δ*orgC flhD::Tn10* | |
| Strain, strain background (*Salmonella enterica* serovar Typhimurium) | SB3272 | This study | Δ*invJ invA::kan flhD::Tn10* | |
| Strain, strain background (*Salmonella enterica* serovar Typhimurium) | SB3273 | This study | Δ*invJ invG::kan flhD::Tn10* | |
| Strain, strain background (*Salmonella enterica* serovar Typhimurium) | SB3274 | This study | Δ*invJ* Δ*prgI flhD::Tn10* | |
| Strain, strain background (*Salmonella enterica* serovar Typhimurium) | SB3289 | This study | Δ*invJ* Δ*orgC* Δ*prgI flhD::Tn10* | |
| Antibody | M2 | Sigma | | Mouse monoclonal antibody to the FLAG epitope |

## Strains and plasmids

All *Salmonella enterica* serovar Typhimurium strains used in this study are derived from the strain SL1344 (*Hoiseth and Stocker, 1981*) and are listed in *Supplementary file 2*. Genetic modifications were introduced into *S.* Typhimurium by allelic exchange using R6K suicide vectors or phage transduction as previously described (*Kaniga et al., 1994*). All plasmids used in these studies were constructed using standard recombinant DNA techniques and are listed in *Supplementary file 3*.

## Bacterial clumping assay

The clumping assay was carried out as previously described (*Lefebre and Galán, 2014*). The background strain used in this assay had ΔinvJ (leading to the production of long needles) and ΔflhD (eliminating motility and thus enhancing clumping) mutations, and carried a plasmid with arabinose-inducible *hilA* to ensure homogenous expression of the T3SS. For exogenous complementation of the ΔorgC mutant strain with OrgC proteins, purified OrgC (60 μg per ml culture) was added at the beginning of the growth period.

## Purification of the Needle Complex

The *S.* Typhimurium SPI-1 encoded T3SS needle complex was isolated as previously described (*Kubori et al., 1998*; *Marlovits et al., 2006*).

## Bacterial two-hybrid assay

Protein-protein interactions by the bacterial two-hybrid assay were carried out as previously described (*Karimova et al., 1998*; *Akeda and Galán, 2004*). Briefly, *E. coli* DHP1 strains transformed with two plasmids expressing the indicated proteins fused to divided adenylate cyclase were pre-cultured in LB broth containing antibiotics and 0.5 mM IPTG at 30°C overnight. The culture was spotted onto LB plates containing X-Gal (40 ug/ml), 1 mM IPTG and antibiotics, and the plates were incubated at 30°C for 16 hr to develop the color indicator. For this assay, a mutant of PrgI lacking its last five amino acids was used as this mutant has been shown to be soluble and unable to polymerize (*Poyraz et al., 2010*), a requirement to be suitable for the bacterial two hybrid assay.

## Pull down assay

*S.* Typhimurium ΔorgC strain carrying pWSK129-based plasmids (*Wang and Kushner, 1991*) expressing either maltose-binding protein (MBP) or MBP-OrgC were grown overnight and cultures were lysed in 20 mM Tris (pH7.4) and 150 mM NaCl by a One Shot (Constant Systems Ltd) cell disruption system. MBP or MBP-OrgC in the cleared lysate was bound to amylose resin, washed and eluted by 10 mM maltose in the same buffer. The final samples were concentrated by TCA precipitation and analyzed by SDS-PAGE followed by immunoblot.

## EM analyses

Bacterial cells or purified NCs were applied to glow-discharged carbon-coated Cu grids and stained with 2% uranyl acetate. Images were acquired with Digital Micrograph (Gatan Inc.) on a FEI Tecnai T12 electron microscope (120 kV) equipped with a US4000 CCD camera (Gatan Inc.).

## Expression and purification of recombinant OrgC proteins from *E. coli*

Unless noted, all *orgC* construct (OrgC wild type or mutants carrying a 6 x His or FLAG epitope tag at their amino termini) were cloned in a pET15 vector so that they could be expressed under a T7 promoter, and the resulting plasmid were transformed into *E. coli* BL21(DE3). Bacterial cultures were grown at 37°C in LB broth containing ampicillin until an $OD_{600}$ of 0.6–0.8 and expression of OrgC was induced by adding 0.15 mM IPTG and subsequently grown overnight at 25°C (to maximize protein solubility). Bacterial cells were collected by centrifugation, resuspended in a buffer containing 20 mM Tris (pH7.4), 150 mM NaCl, 10 mM imidazole, 10 μg/ml DNase, 0.5 mM MgSO4 and a protease inhibitor cocktail (complete EDTA-free, Roche), and lysed in a One Shot (Constant Systems Ltd) cell disruption system. The clarified cell lysate was bound to Ni-NTA agarose resin (Qiagen), washed in 20 mM Tris (pH7.4), 150 mM NaCl, and eluted in elution buffer [20 mM Tris (pH7.4), 150 mM NaCl and 250 mM imidazole]. The eluted OrgC protein was further purified by ion-exchange chromatography (Hi-trap Q, GE Health Life Sciences) in 20 mM Tris (pH 8.0) using a gradient of 0–1M

NaCl. The OrgC containing fractions were applied to a size exchange chromatography column (Superdex 75, GE Health Life Sciences) in 20 mM Tris (pH 7.4) and 150 mM NaCl. The OrgC protein eluted as a single peak. OrgC tagged with MBP at the N-terminus was constructed in the pMAL-c5x vector (NEB), transformed into *E. coli* BL21(DE3), grown in LB broth at 37 °C until an $OD_{600}$ of 0.6–0.7, when expression of MBP-OrgC was induced by addition of 0.3 mM IPTG and further incubation at 37 °C for 4.5 hs. Bacterial cells were disrupted as indicated above and MBP-OrgC was isolated using amylose resin (NEB) following the manufacture's instruction, and further purified by ion-exchange and size-exclusion chromatography as described above.

## Dynamic light scattering (DLS)

DLS was used to monitor the polymerization of PrgI needles in vitro following the method described by Poyraz et al. (*Poyraz et al., 2010*). DLS data was acquired using a PD2000 DLS instrument (Precision Detectors, Bellingham, MA) and analyzed using the manufacturer's software (Precision Deconvolve Version 4.5). The DLS acquisition parameters used were 5 us and 30 us sampling times; four repetitions; and 60 scans per repetition. For DLS, a PrgI double mutant ($PrgI^{V65A/V67A}$), which was used by Poyraz et al. (*Poyraz et al., 2010*) in the DLS assay, was expressed and purified as a fusion protein (His-tag-GB1-$PrgI^{V65A/V67A}$) similar to the expression and purification of His-tag-GB1-OrgC described below. After nickel affinity chromatography and digestion with TEV protease (to remove the tag), $PrgI^{V65A/V67A}$ was dialyzed in DLS buffer (20 mM HEPES pH 7.5, 50 mM NaCl) and concentrated to about 4 mg/mL. Full-length recombinant OrgC was expressed and purified by nickel affinity chromatography as described above and dialyzed in DLS buffer. Samples for DLS typically contained 300 μL of 4 mg/mL PrgI solution in DLS buffer $PrgI^{V65A/V67A}$ (*Poyraz et al., 2010*). Prior to the DLS experiment, protein samples were passed through 0.45 μm syringe filter, and DLS data was acquired at 30 min interval from 0 to 180 min.

## Protein NMR spectroscopy

To express OrgC for NMR experiments, a deletion construct of the first N-terminal 21 amino acids was sub-cloned into pDZ1 (*Zhong et al., 2012*) to create a fusion protein (His-tag-GB1-OrgC) consisting of the N-terminal deletion of OrgC, a His-tag for protein purification, the B1 immunoglobulin-binding domain of Streptococcus protein G (GB1) as solubility enhancer, and a tobacco etch virus (TEV) protease cleavage site. PrgI was expressed and purified for NMR studies as previously described (*Wang et al., 2007*). For protein expression, plasmid expressing His-tag-GB1-OrgC or PrgI was transformed in *E. coli* BL21(DE3). OrgC and PrgI uniformly labeled with $^{15}N$ were obtained by cell growth in 1L M9 minimal media supplemented with $^{15}NH_4Cl$, antibiotics, trace minerals (ATCC MD-TMS), and vitamins (ATCC MD-VS). Unlabeled OrgC and PrgI were obtained by cell growth in 1L LB media. Typically, cells were grown to $OD_{600}$ ~0.8 at 37°C, induced with 1 mM IPTG, and protein expression was continued by cell growth at 15°C. Cells were lysed by sonication and the recombinant protein was purified by nickel affinity chromatography as described (*McShan et al., 2016*). The His-tag-GB1-OrgC fusion protein was digested with TEV protease, followed by a second round of nickel affinity chromatography to yield recombinant OrgC.

The assignment of the NMR peaks of PrgI has been reported previously (*Wang et al., 2007*), and the NMR assignments for OrgC followed essentially the methods used in assigning PrgI (*Wang et al., 2007*). Purified $^{15}N$-labeled OrgC and unlabeled PrgI were dialyzed in NMR buffer (20 mM sodium phosphate, 20 mM NaCl, pH 7), and several samples of $^{15}N$ OrgC:PrgI complexes at varying molar ratios were prepared. Likewise, $^{15}N$ PrgI was titrated with unlabeled OrgC in a similar manner. The titration was monitored by acquiring 2D $^1H$-$^{15}N$ heteronuclear single quantum coherence (HSQC) spectra for each titration point at room temperature using a Bruker Avance 800 MHz spectrometer equipped with a cryoprobe. The NMR spectra were processed by NMRPipe (*Delaglio et al., 1995*) and analyzed by NMRView (*Johnson, 2004*). Weighted chemical shift deviations were calculated and plotted as described (*McShan et al., 2016*).

## NMR structure determination of OrgC

NMR samples for structure determination consisted of 500 μL volume of 1.2 mM $^{15}N$- or $^{15}N/^{13}C$-labeled OrgC (residues 21–150) in 2 mM 2-(N-morpholino)ethanesulfonic acid buffer (pH 6.5) and 75 mM NaCl. Typical NMR samples were dissolved in 10% $D_2O$, however, for 3D $^{13}C$-edited HMQC

NOESY (see below), the NMR sample was lyophilized, and re-dissolved in 100% $D_2O$. NMR spectra were acquired at 25°C using a Bruker Avance 800 MHz NMR equipped with a triple-resonance cryo-probe or a Bruker Avance 600 MHz NMR equipped with a room-temperature triple resonance probe. NMR data were processed with NMRPipe (*Delaglio et al., 1995*) and analyzed with NMRView (*Johnson, 2004*). NMR peak assignments were obtained from 2D $^{15}$N HSQC (*Grzesiek and Bax, 1993a*) and 3D HNCACB (*Wittekind and Mueller, 1993*), HNCA (*Grzesiek et al., 1992*), CACBCONH (*Grzesiek et al., 1992*), and HBHA(CO)NH (*Grzesiek and Bax, 1993b*). Analysis of the $^{13}$Cα, $^{13}$Cβ and $^{1}$Hα secondary chemical shifts (*Wishart and Nip, 1998*) identified the helical regions of OrgC. For structure calculations, distance restraints were obtained from NOEs (nuclear Overhauser effect) assigned from 3D $^{15}$N-edited HSQC NOESY ($t_{mix}$120 ms) and 3D $^{13}$C-edited HMQC NOESY ($t_{mix}$120 ms). Using intra-residue NOEs as internal standards, the NOEs were categorized based on peak intensities into strong, medium, and weak NOEs, and assigned upper distance limits of 2.7 Å, 3.5 Å, and 5.5 Å, respectively. Dihedral angle restraints of phi $-60 \pm 20°$ and psi $-40 \pm 20°$ were used for alpha helical regions identified by the secondary chemical shifts. Structure calculations were done using CYANA (*Güntert, 2004*), typically, as follows: 170 structures were calculated by simulated annealing and molecular dynamics followed by energy minimization of 100 structures with the lowest energies. The 20 lowest energy structures were analyzed to identify violations in distance and dihedral restraints. Structure calculations were repeated iteratively by addition of newly identified NOEs until the global fold of the protein emerged. The 40 CYANA structures with the lowest target functions were used as starting models for further structural calculations and refinement using AMBER7 (*Case, 2002*) following a similar protocol described earlier (*Dames et al., 2002*). Briefly, the AMBER7 structure calculation was initiated with 1000 steps of energy minimization, followed by simulated annealing *in vacuo* for 20 ps at 1000 K, a second round of simulated annealing for 20 ps at 500 K, and a final energy minimization of 1000 steps. In the final stages of the structural refinements, AMBER7 with the generalized Born potential (*Xia et al., 2002*) was used as implicit water model in NMR structure calculations. The twenty lowest energy structures were used for structural analysis using Pymol. The 20 PDB coordinates, the assigned chemical shifts, and the restraints used in the NMR structure determination were deposited at the RCSB Protein Data Bank with the accession code PDB ID 6CJD and BMRB ID 30417. The above data were used to generate *Figure 7*, *Figure 6—figure supplement 1*, *Figure 7—figure supplements 1*, *2*, *3* and *4*, and *Supplementary file 1*.

## Acknowledgements

We thank members of the Galán laboratory for critical review of this manuscript. J S was supported by a fellowship from the Pew Latin American Fellows Program in the Biomedical Sciences. This work was supported by NIH grants AI079022 (to JEG), AI074856 (to R N D), and P20GM103418 (to MCW).

## Additional information

### Funding

| Funder | Grant reference number | Author |
|---|---|---|
| National Institutes of Health | AI079022 | Jorge E Galan |
| National Institutes of Health | AI074856 | Roberto De Guzman |
| National Institutes of Health | P20GM103418 | Mason Wilkinson |

The funders had no role in study design, data collection and interpretation, or the decision to submit the work for publication.

### Author contributions

Junya Kato, Conceptualization, Data curation, Formal analysis, Investigation, Methodology, Writing—original draft, Writing—review and editing; Supratim Dey, Data curation, Formal analysis, Investigation; Jose E Soto, Formal analysis, Investigation, Methodology, Writing—review and editing;

Carmen Butan, Formal analysis, Investigation, Methodology; Mason C Wilkinson, Investigation, Methodology; Roberto N De Guzman, Data curation, Formal analysis, Supervision, Funding acquisition, Investigation, Methodology, Project administration, Writing—review and editing; Jorge E Galan, Conceptualization, Formal analysis, Supervision, Funding acquisition, Writing—original draft, Project administration, Writing—review and editing

### Author ORCIDs
Jorge E Galan  http://orcid.org/0000-0002-6531-0355

### Decision letter and Author response
Decision letter https://doi.org/10.7554/eLife.35886.028
Author response https://doi.org/10.7554/eLife.35886.029

## Additional files

### Supplementary files
• Supplementary file 1. Restraints and structural statistics for 20 NMR structures of OrgC.
DOI: https://doi.org/10.7554/eLife.35886.019

• Supplementary file 2. Bacterial strains used in this study.
DOI: https://doi.org/10.7554/eLife.35886.020

• Supplementary file 3. Plasmids used in this study.
DOI: https://doi.org/10.7554/eLife.35886.021

• Transparent reporting form
DOI: https://doi.org/10.7554/eLife.35886.022

### Data availability
The 20 PDB coordinates, the assigned chemical shifts, and the restraints used in the NMR structure determination were deposited at the RCSB Protein Data Bank with the accession code PDB ID 6CJD and BMRB ID 30417. The above data were used to generate Fig. 7, Figure 6-figure supplement 1, Figure 7-figure supplement 1, 2, 3, and 4, and Supplementary File 1.

The following datasets were generated:

| Author(s) | Year | Dataset title | Dataset URL | Database and Identifier |
|---|---|---|---|---|
| Kato J, Dey S, Soto JE, Butan C, Wilkinson MC, De Guzman RN, Galán JE | 2018 | NMR NMR structure determination of OrgC | http://www.rcsb.org/pdb/search/structid-Search.do?structureId=6CJD | RCSB Protein Data Bank, 6CJD |
| Kato J, Dey S, Soto JE, Butan C, Wilkinson MC, De Guzman RN, Galán JE | 2018 | NMR NMR structure determination of OrgC | http://www.bmrb.wisc.edu/data_library/summary/index.php?bmrbId=30417 | Biological Magnetic Resonance Data Bank, 30417 |

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
