## [Decision Letter]

Thank you for submitting your article "A protein secreted by the *Salmonella* type III secretion system controls needle filament assembly" for consideration by *eLife*. Your article has been reviewed by Gisela Storz as the Senior Editor, a Reviewing Editor, and three reviewers. The reviewers have opted to remain anonymous.

The reviewers have discussed the reviews with one another and the Reviewing Editor has drafted this decision to help you prepare a revised submission.

Summary:

Many Gram-negative pathogen employ a syringe-like secretion system – the injectisome – to secrete effector proteins into host cells. The injectisome is evolutionary related to bacterial flagella. A type III secretion system (T3SS) powers protein export via both nanomachines. In the present manuscript, Kato et al. investigated the molecular function of a hitherto poorly characterized component of the *Salmonella* SPI-1 injectisome, OrgC. The authors found that OrgC is secreted in a T3SS-dependent manner and is required for efficient needle assembly, but not for needle elongation. In a clever set of experiments, the authors show that extracellular addition of recombinant OrgC is able to complement the needle assembly defect of a ∆*orgC* mutant strain. Finally, the authors determined that OrgC interacts with the injectisome needle protein PrgI and solved the OrgC structure. The injectisome is a macromolecular machine of remarkable complexity. How bacteria manage to assemble such complex structures remains a poorly understood and is of interest to a general audience. The work is elegantly carried out and the manuscript is well written.

Essential revisions:

1) The authors should provide further evidence of OrgC-dependent in initiation of PrgI needle polymerization, either in vitro (by purifying OrgC-His and PrgI and assaying for filament assembly using electron microscopy or fluorescence as readouts) or in vivo (by monitoring needle polymerization in vivo by addition of recombinant OrgC and/or PrgI* in a ∆*orgC* ∆*prgI* double mutant).

2) The authors should determine if plasmid overexpression of OrgC is having an effect on expression or secretion. For example, is overexpression affecting the regulation of HilA?

3) It is obvious that purified OrgC has an effect when added exogenously. This is shown by the indirect "clumping" assay. However, it would be more convincing if the authors repeated the experiment in Figure 2E by analyzing the needle complexes from the wild type and orgC mutant after ectopic addition of purified OrgC since this assay is more direct.

4) Figure 6FG: The phenotype of the orgC deletion mutants is not informative without showing that these mutants are stable and still secreted.

5) The authors need to provide additional quantification and/or information for the following:

-Figure 1CD: Improve figure labels and add a legend. (Currently, the labeling of the figure is misleading. The authors study the secretion of OrgC-3xFLAG, which is expressed from a plasmid, not of endogenous orgC. Does expression of OrgC-3xFLAG from a plasmid impact expression/function of the SPI1 T3SS?)

-Figure 2: Needs quantification and statistical analyses. (PrgI secretion in a ∆*invJ* ∆*orgC* double mutant versus a ∆*invJ* mutant would be an appropriate control.)

-Figure 3AB: Clarify what *Salmonella* strain was analyzed in Figure 3A and how the two proteins OrgC and OrgC deltaN21 were detected in the Western blot experiment. The experiment in Figure 3B deserves a complete description and labeling of the figure. (A secreted protein control and a separate control for a *Salmonella* strain that cannot perform T3SS is needed to document that the amounts of MBP and MBP-OrgC in the culture supernatant are due to cell lysis.)

-Figure 4E: Needs quantification and statistical analyses to support claim that purified His-OrgC restores SipB secretion in the orgC mutant strain as compared to wild-type.

-Figure 5: Provide more information about the PrgI^∆C5^ mutant (not mentioned in the text or figure legend).

-Figure 6FG: The authors should report the phenotype of the PrgI point mutations predicted to not interact with OrgC. The statement that the PrgI point mutations did not alter needle assembly is contradictory to the conclusion that the absence of OrgC results in a needle assemble defect.

-Add list with strain and plasmid genotypes.

---

## [Author Response]

Essential revisions:1) The authors should provide further evidence of OrgC-dependent in initiation of PrgI needle polymerization, either in vitro (by purifying OrgC-His and PrgI and assaying for filament assembly using electron microscopy or fluorescence as readouts) or in vivo (by monitoring needle polymerization in vivo by addition of recombinant OrgC and/or PrgI* in a ∆orgC ∆prgI double mutant).

As suggested by the reviewers, we have now included the results of experiments conducted to assess the role of OrgC in the initiation of needle polymerization. More specifically, we have used dynamic lights scattering to monitor needle protein polymerization in vitro and as shown in the revised Figure 5D, in the presence of OrgC, PrgI needle polymerization is significantly accelerated, which is consistent with our model.

2) The authors should determine if plasmid overexpression of OrgC is having an effect on expression or secretion. For example, is overexpression affecting the regulation of HilA?

We have carried out the requested experiment, which is now shown in the modified Figure 2A. As shown, neither the absence (∆*orgC*) nor the expression of OrgC from the low copy plasmid pWSK129 has any effect on the expression of the secreted proteins as indicated by the western blot analysis.

3) It is obvious that purified OrgC has an effect when added exogenously. This is shown by the indirect "clumping" assay. However, it would be more convincing if the authors repeated the experiment in Figure 2E by analyzing the needle complexes from the wild type and orgC mutant after ectopic addition of purified OrgC since this assay is more direct.

The results of the requested experiments are shown in the modified Figure 4H. Although, under the conditions tested, wild-type levels of fully assembled needle complexes were not entirely recovered after addition of purified OrgC, we detected a clear increase in the percentage of NC bases with needle substructures compared to the ∆*orgC* strain in the absence of exogenous addition of OrgC.

4) Figure 6FG: The phenotype of the orgC deletion mutants is not informative without showing that these mutants are stable and still secreted.

We have now shown in Figure 6—figure supplement 3 that the OrgC deletion mutants are expressed and secreted at wild type levels.

5) The authors need to provide additional quantification and/or information for the following:-Figure 1CD: Improve figure labels and add a legend. (Currently, the labeling of the figure is misleading. The authors study the secretion of OrgC-3xFLAG, which is expressed from a plasmid, not of endogenous orgC. Does expression of OrgC-3xFLAG from a plasmid impact expression/function of the SPI1 T3SS?)

We regret the misunderstanding. The OrgC-FLAG proteins are expressed from the chromosome (there is no overexpression). In any case, expression of OrgC from a plasmid does affect T3SS as shown in complementation experiments depicted in Figure 2A.

-Figure 2: Needs quantification and statistical analyses. (PrgI secretion in a ∆invJ ∆orgC double mutant versus a ∆invJ mutant would be an appropriate control.)

We have provided the quantification of Figure 2 in the revised figure (see additional panel). In addition, in Figure 2—figure supplement 1 we show the secretion profile of PrgI in wild type, ∆*invJ* ∆*orgC,* and ∆*invJ S. Typhimurium* strains.

-Figure 3AB: Clarify what Salmonella strain was analyzed in Figure 3A and how the two proteins OrgC and OrgC deltaN21 were detected in the Western blot experiment. The experiment in Figure 3B deserves a complete description and labeling of the figure. (A secreted protein control and a separate control for a Salmonella strain that cannot perform T3SS is needed to document that the amounts of MBP and MBP-OrgC in the culture supernatant are due to cell lysis.)

The secretion profile of a T3SS defective strain (∆*invA* mutant) and a secreted protein control (translocase SipB) for all the analyzed strains have now been included in a new panel (Figure 3B).

-Figure 4E: Needs quantification and statistical analyses to support claim that purified His-OrgC restores SipB secretion in the orgC mutant strain as compared to wild-type.

We have provided the quantification for InvJ, which is the most robust marker. The secretion phenotype for the other proteins, while detectable, is not sufficiently robust to achieve significance. We have altered the text to note this point.

-Figure 5: Provide more information about the PrgI^∆C5^ mutant (not mentioned in the text or figure legend).

The requested information has now been provided.

-Figure 6FG: The authors should report the phenotype of the PrgI point mutations predicted to not interact with OrgC. The statement that the PrgI point mutations did not alter needle assembly is contradictory to the conclusion that the absence of OrgC results in a needle assemble defect.

The point of the experiment in Figure 6G was to show that the PrgI mutations did not affect needle assembly since defects in needle assembly would indirectly affect the secretion of other proteins. The secretion phenotype of the ∆*orgC* mutant is not nearly as robust as the clumping phenotype. As shown in Figure 6G, the phenotype of the PrgI mutations in the clumping assay was intermediate in comparison with the phenotype of the ∆*orgC* mutant. This is expected given the complexity of the interface between OrgC and PrgI, which cannot be easily altered by mutagenesis without compromising needle assembly. Consequently, the PrgI mutations, while exhibiting a phenotype in the more robust clumping assay, they do not show a measurable phenotype in the secretion assay. We have clarified this point in the text.

-Add list with strain and plasmid genotypes.

The strains and plasmids have now been added in a Supplementary file 2 and Supplementary file 3, which we had originally accidently did not upload.